# Current Methods in the Study of Nanomaterials for Bone Regeneration

**DOI:** 10.3390/nano12071195

**Published:** 2022-04-02

**Authors:** Manabu Tanaka, Makoto Izumiya, Hisao Haniu, Katsuya Ueda, Chuang Ma, Koki Ueshiba, Hirokazu Ideta, Atsushi Sobajima, Shigeharu Uchiyama, Jun Takahashi, Naoto Saito

**Affiliations:** 1Department of Orthopedic Surgery, Okaya City Hospital, 4-11-33 Honcho, Okaya, Nagano 394-8512, Japan; sigeuti@shinshu-u.ac.jp; 2Institute for Biomedical Sciences, Interdisciplinary Cluster for Cutting Edge Research, Shinshu University, 3-1-1 Asahi, Matsumoto, Nagano 390-8621, Japan; 21hb401k@shinshu-u.ac.jp (M.I.); 19hb402j@shinshu-u.ac.jp (K.U.); 20hb403b@shinshu-u.ac.jp (C.M.); 21bs201b@shinshu-u.ac.jp (K.U.); saitoko@shinshu-u.ac.jp (N.S.); 3Biomedical Engineering Division, Graduate School of Medicine, Science and Technology, Shinshu University, 3-1-1 Asahi, Matsumoto, Nagano 390-8621, Japan; ideta@shinshu-u.ac.jp; 4Biomedical Engineering Division, Graduate School of Science and Technology, Shinshu University, 3-1-1 Asahi, Matsumoto, Nagano 390-8621, Japan; 5Department of Orthopedic Surgery, Shinshu University School of Medicine, 3-1-1 Asahi, Matsumoto, Nagano 390-8621, Japan; soba@shinshu-u.ac.jp (A.S.); jtaka@shinshu-u.ac.jp (J.T.); 6Department of Orthopedics (Lower Limbs), Social Medical Care Corporation Hosei-kai Marunouchi Hospital, 1-7-45 Nagisa, Matsumoto, Nagano 390-8601, Japan

**Keywords:** nanomaterials, bone regeneration, osteoblast, biomaterials, scaffolds, culture, evaluation, in vitro, in vivo

## Abstract

Nanomaterials show great promise as bone regeneration materials. They can be used as fillers to strengthen bone regeneration scaffolds, or employed in their natural form as carriers for drug delivery systems. A variety of experiments have been conducted to evaluate the osteogenic potential of bone regeneration materials. In vivo, such materials are commonly tested in animal bone defect models to assess their bone regeneration potential. From an ethical standpoint, however, animal experiments should be minimized. A standardized in vitro strategy for this purpose is desirable, but at present, the results of studies conducted under a wide variety of conditions have all been evaluated equally. This review will first briefly introduce several bone regeneration reports on nanomaterials and the nanosize-derived caveats of evaluations in such studies. Then, experimental techniques (in vivo and in vitro), types of cells, culture media, fetal bovine serum, and additives will be described, with specific examples of the risks of various culture conditions leading to erroneous conclusions in biomaterial analysis. We hope that this review will create a better understanding of the evaluation of biomaterials, including nanomaterials for bone regeneration, and lead to the development of versatile assessment methods that can be widely used in biomaterial development.

## 1. Introduction

Bone regeneration research has a long and diverse history. In vivo and in vitro exposure experiments are two of the most widely used experimental tools employed to evaluate bone formation. In the former, bone defects and fractures are artificially created in test animals, such as rodents, for the assessment of various treatment materials and methods [1,2]. In vitro experiments have the advantage of providing a more detailed picture of the process by which a substance exerts its effects on cells and tissues in a target organ under specific conditions. They include the exposure of cultured cells to a drug [3] or culturing cells on a material [4] to understand the proliferation, morphology, and intracellular chemistry of cells associated with bone regeneration.

Each organ in the human body functions in relation to others through endocrine and autonomic controls. When a bone is subject to a fracture or substances used in fracture treatment, the other organs are also affected indirectly. Moreover, secreted osteogenic signals may activate or inhibit other forms of signaling. In vivo experiments show how all substances involved in osteogenesis impact treatment outcome, although it is often difficult to pinpoint which substances exert which effects. Of different but equal concern is that the behavior of cells taken from animals and cultured in vitro may not reflect their original form or function in vivo [5]. Whether in vivo or in vitro, cells and organisms respond differently under different conditions.

In recent years, a number of nanomaterials have been developed as potential materials in bone regenerative medicine [6]. Although most are used as composite materials within bone regeneration scaffolds, there are numerous methods to evaluate their efficacy and safety as raw materials [7] or as composite materials [8]. A wide variety of animals, cells, and culture media are employed for this purpose [9,10]. As the clinical application of nanomaterials approaches, it will be necessary to unify the experimental methods published to date to objectively compare the nanomaterials available. As we previously reported [11], the results of cell proliferation, cell differentiation, and other functions differ depending on the conditions of the culture medium. Similarly, the effect of nanomaterials on bone formation may also vary in relation to the experimental set-up.

The purpose of this review is to evaluate and compare the current experimental systems related to the development of biomaterials for bone regeneration, with a particular focus on nanomaterials in the in vitro setting. We will cover the main evaluation conditions that are widely used for the development of nanomaterials and other biomaterials, especially towards reducing the need for animal experiments. This review will provide the reader with the necessary considerations and concepts to use as a basis for effective research in the evaluation of nanomaterials as well as all biomaterials for bone regeneration by explaining the importance of establishing appropriate experimental conditions in accordance with the objectives and goals of the study.

## 2. Promising Nanomaterials for Bone Regeneration Materials

Nanomaterials are particles of 100 nm in size or less. Specifically, they are defined as a natural, incidental, or manufactured material containing particles in an unbound, aggregate, or agglomerate state, in which 50% or more of the particles have one or more external dimensions in the size range of 1–100 nm [12]. Nanomaterials include titanium dioxide (TiO_2_), calcium carbonate, silica, carbon black, zinc oxide, graphene oxide, carbon nanotubes (CNTs), carbon nanohorns (CNHs), fullerenes, and more. They have a wide range of applications involving cosmetics, catalysts, quantum dots, contrast agents, drug delivery, electronic device junctions, and wiring materials. Nanomaterials are also blended as fillers in numerous materials, such as rubber, resin, paper, ink, paint, and sealant, to produce various physical properties.

Nanomaterials have excellent mechanical properties due to their volumetric, surface, and quantum effects. For example, the compressive and flexural strengths of cement mortars with nano-SiO_2_ or nano-Fe_2_O_3_ are higher than those of a blank group [13]. Nano-Al_2_O_3_ ceramics also have higher flexural strength as compared with microscale monolithic alumina ceramics [14]. When nanoparticles are added to common materials, the particles refine the material grain to some extent and form intra- or intergranular structures, thereby improving grain boundaries and promoting the mechanical properties of the substance [15].

Surface properties are a major reason why materials behave differently at the nanoscale. For instance, surface functionalization by nanomaterials enables the improvement and addition of properties useful for medical applications. The modification of nanomaterial surfaces can be achieved by two different approaches: non-covalent and covalent. Non-covalent bonding is a weak interaction that is often used with metals and silica [16]. The method is relatively simple but susceptible to such variables as pH and ionic strength [17]. In contrast, the covalent attachment of ligands to nanomaterial surfaces is performed using linker molecules. One example is polyethylene glycol (PEG), which can be synthesized with a specific functional group and used as a linker to enable a wide range of functions. Nanomaterials can even be modified at several sites to produce multiple functionalization [18]. Recently, the use of sensitive binding, including pH-sensitivity and thermo-sensitivity, to develop nanoplatforms for controlled drug release has been investigated [19].

Each nanomaterial has its own advantages and drawbacks (Table 1). Ceramics are similar to bone components and are believed to have osteoinductive ability and biocompatibility, although the cytotoxicity of nano-sized ceramics when taken up by cells is often a concern. Polymers are highly bioabsorbable and regarded as relatively safe materials. They are also easy to process and can be formed into nanofibers to create scaffolds. On the other hand, the rapid absorption of polymers may lead to a loss of strength and an over-release of drugs. Carbon possesses excellent strength and conductivity and is believed to transmit bioelectric signals. However, it is not bioabsorbable, and its long-term biological safety remains controversial [20]. Gold nanoparticles have excellent biocompatibility and photothermal stability, and can absorb near-infrared waves. On the other hand, they are not bioabsorbable, and their long-term safety is unknown. Titanium-based nanomaterials exhibit high load-bearing strength and biocompatibility, but are not bioresorbable and have a slow biological response and low antimicrobial resistance [21]. Lastly, liposomes possess excellent drug-retention ability [22], but are unsuitable for forming scaffolds on their own, since they cannot be processed into fibrous forms, such as polymers.

### 2.1. Nanomaterials as Fillers in Bone Regeneration Scaffolds

Fracture healing is described in four phases (Figure 1). The first inflammatory phase occurs 1–5 days immediately following the fracture, during which a hematoma is formed at the fracture site and the hypoxic microenvironment limits oxygen and nutrient supply. Local inflammation activates cell migration. The second stage features the formation of fibrocartilaginous callus, and the third stage is fibrocartilaginous callus replacement by a bony callus. In the fourth stage, the bony callus is replaced by hard cortical bone, and the fracture healing process is complete [23]. Various cytokines are secreted in fracture healing, including those for osteoblast differentiation and angiogenesis.

Owing to their unique biocompatibility, drug delivery, and structural characteristics, nanomaterials have been extensively tested as candidates for bone regeneration materials (Figure 2).

Scaffolding materials are an essential component of bone regeneration. Bone has a high Young’s modulus (E = 18.6 GPa [24]) and is one of the hardest load-bearing tissues in the human body. The primary function of scaffolding materials is to provide a suitable environment for cell growth and temporary mechanical support to the defect site [10]. Ideally, scaffolds should have the following properties: (1) three-dimensional, porous, bioconductive, biocompatible, and bioresorbable; (2) mechanical properties comparable to those of bone tissue; and (3) surface properties suitable for promoting cell attachment, proliferation, and differentiation [25].

Bioabsorbable polymers exhibit several characteristics that make them suitable for bone regeneration, but lack the necessary mechanical strength needed for an ideal bone tissue scaffold. The addition of nanomaterials as fillers to such matrix polymers can enhance their mechanical properties [26]. Accordingly, a number of nanomaterials have been incorporated into bioresorbable polymers as scaffold fillers and studied for their effects on bone regeneration. For example, ceramic nanomaterials such as hydroxyapatite (HA) and bioactive glasses, carbon-based materials including CNTs and graphene, and metal-based nanoparticles (NPs) such as gold and TiO_2_ have been widely tested as potential scaffold fillers [27,28,29,30,31,32] (Table 2). In addition to their role in strengthening the mechanical properties of scaffolds, nanomaterials may also have specific effects on cell signaling pathways, such as increasing protein adhesion to the scaffold [33].

HA is a major inorganic component of bone tissue. Therefore, hydroxyapatite nanoparticles (HANPs) are very attractive biomaterials in scaffold design due to their ability to mimic the natural inorganic phase of bone as compared with organic polymer NPs. The addition of HANPs as a filler to matrix materials to construct composite scaffolds has been intensively studied as an approach to augment osteogenesis. Polymers such as chitosan, gelatin, polycaprolactone, polylactic and glycolic acids, and PEG are among the matrix materials used in combination with HANPs [6].

The use of carbon nanofibers produced by electrospinning, which is an effective method of creating nano-sized fibers by applying high voltage to a needle tip containing a polymer solution, has been investigated in the medical field, and its use as a scaffold for bone regeneration therapy has attracted attention [34]. When thin carbon nanofiber web implants containing recombinant human bone morphogenetic protein-2 (BMP-2) were implanted into the lumbar fascia of ddY mice, the formation of bone matrix and bone marrow structures around the implants was observed 3 weeks later. Although carbon fiber is a non-biodegradable material, nano-sized carbon fibers may conceivably be incorporated into the bone matrix as a filler to improve mechanical strength.

### 2.2. Nanomaterials Used in Drug Delivery Systems (DDS)

NPs can easily bypass the cell membrane barrier by endocytosis [35]. The mechanisms of particle internalization (endocytosis) include phagocytosis (cell “eating”) and pinocytosis (cell “drinking”). Phagocytosis generally occurs for microparticles, while pinocytosis occurs for NPs. This particular feature of NPs is being explored for possible use in DDSs for side effect suppression and sustained release. NPs incorporated into cells are known to induce autophagy. Autophagy is a major cell survival-promoting mechanism that is activated under cellular stress conditions. Autophagy promotes the circulation of cellular components and provides energy to starving cells while simultaneously removing damaged proteins and organelles. Such functions are essential for osteocyte differentiation and immune cell polarization and are thought to play a central role in bone regeneration [36] (Figure 3).

BMP-2 is a potent osteogenic factor for the induction of bone formation [37]. It is a multi-faceted protein, and when administered systemically, can cause such side effects as life-threatening events, implant displacement, and infection [38]. As an alternative to systemic BMP-2 administration, several scaffolds and NP systems have been developed as DDS of BMP-2 [39].

VEGF is another important growth factor promoting bone healing and regeneration. VEGF also plays a prominent role in osteoclastogenesis by influencing the bone remodeling process, which is the final step in bone healing. VEGF-containing NPs have been reported to promote angiogenesis [40] and repair myocardial infarction [41]. The short half-life of VEGF suggests that when successfully administered at optimal doses, it can improve angiogenesis and regeneration in fractured bone [42].

Our group has been developing a DDS with CNHs for the local treatment of metastatic bone destruction. We used calcium phosphate (CaP) as a loading mediator for the bone resorption inhibitor ibandronate (IBN) and observed a significant effect of CNH-CaP-IBN on bone resorption [43]. We are currently evaluating the efficacy of our DDS in vivo to determine if it not only inhibits bone resorption, but also promotes bone formation in the fractured area by means of CNHs. As well as CNHs, various NPs have been studied as media for DDS, including liposomes, polylactic acid, poly lactic-co-glycolic acid, PEG, and silica [22,43,44,45,46] (Table 2).

**Table 2 nanomaterials-12-01195-t002:** Examples of inorganic nanomaterial composites used for bone tissue regeneration.

Application	Nanomaterial	Base	Fabrication Technique	Reference
Filler	Hydroxyapatite nanoparticles	Chitosan	Electrospinning	[27]
	Bioactive glass nanoparticles	Polyethylene glycol dimethacrylate	Bioprinting	[28]
	Carbon nanotubes	Ultrahigh-molecular-weight polyethylene	Thermal compression	[29]
	Graphene oxide	Alginate/gelatin	3D bioprinting	[30]
	Gold nanoparticles	Poly (L-lactic acid)	Electrospinning	[31]
	TiO_2_ nanoparticles	Poly (D, L-lactic acid)	Solvent casting	[32]
Drug delivery	Ibandronate-loaded carbon nanohorns	Calcium phosphate	Coprecipitation	[43]
	Desferrioxamine-loaded liposomes	Hydrogel	Physical blending method	[22]
	Breviscapine-loaded poly (D, L-lactic acid) nanoparticles	-	Spontaneous emulsification solvent diffusion method	[44]
	BMP-2-loaded poly (L-lactic acid)	-	Electrospinning	[45]
	Dexamethasone-loaded mesoporoussilica nanoparticles	Poly (L-lactic acid)/ Poly (ε-caprolactone) nanofibrous scaffold	Thermally induced phase separation	[46]

### 2.3. Bone Regeneration Research Using CNTs

CNTs have excellent mechanical properties, a high aspect ratio, and thermal and electrical conductivity. Thus, they are attractive materials to augment the scaffold properties of bone regeneration materials [47,48].

Multi-walled carbon nanotubes (MWCNTs) can promote the calcification ability of osteoblasts [49] and inhibit osteoclast differentiation and function when exposed to cells at nano-size [50]. Other studies have used MWCNTs as a bulk material or composite material with ultrahigh-molecular-weight polyethylene (UHMWPE) or alumina for bone regeneration scaffolds [29,51,52,53]. The healing of a 5 mm diameter defect was observed in vivo using a mouse cranial bone defect model, and scaffolds made solely of MWCNTs were able to heal the defect better than interconnected porous calcium hydroxyapatite (IP-CHA) scaffold materials. The cell adhesion and proliferation functions of this scaffold were similar to those of IP-CHA, although protein adsorption was superior [52]. Meanwhile, both UHMWPE-CNT and alumina-CNT composites improved the mechanical properties of scaffolds, but did not promote osteogenic function, as seen with single particles [29,53]. The above results indicate that it is difficult to add bioactivity to nanomaterials by simply mixing bioactive nanomaterials with non-bioactive materials. We need to first understand the mechanism of nanomaterial bioactivity, and then consider composites that retain their properties.

## 3. In Vivo Evaluation of Biomaterials for Bone Regeneration

Even if cells are transplanted directly into the body, they often leave the transplant site without providing bone formation activities. Friedenstein et al. demonstrated that bone formation occurred by transplanting bone marrow itself or mesenchymal stem cells grown from bone marrow into the renal membrane, or by transplanting cells subcutaneously into a diffusion chamber, which is a closed space impermeable to cells but permissive to molecules such as growth factor proteins. Thus, bone formation could take place by the transplantation of cells into the body [54], although transplantation into the renal membrane or using a chamber is difficult to apply in common bone diseases. Carriers that can retain cells and growth factors have since been developed in experimental models and are now available for real-world application. Porous HA ceramic, which is also used in clinical practice, is frequently employed as a carrier [51]. Such carriers are able to reproduce bone formation in vivo. A common method to evaluate the ability of bone defect treatment is to create bone defects of non-healing size in rat skulls, implant them with test materials, and analyze the bone healing process [2]. The size of the bone defect is approximately 8 mm in diameter, and the status of bone repair is judged histologically 4–12 weeks after surgery. A model that creates a bone defect in the diaphysis of the femur has also been considered [55], but it is technically difficult to reproduce in rats and other small mammals. Since experiments using large mammals such as sheep are not feasible in all laboratories, this system is unlikely to become a universal standard. Animal trials are necessary to elucidate biological processes and test new therapies towards improving human health. However, for ethical reasons, animals should be properly managed, and only a minimal number should be used. The “3Rs” (replacement, reduction, and refinement) are highlighted to guide researchers and ethics committees in the humane experimentation of animals in science and are enshrined in the laws governing animal use in many countries [56].

## 4. In Vitro Evaluation of Biomaterials for Bone Regeneration

The need for in vitro evaluation methods is increasing due to the reproducibility problems of animal experiments that accompany the ethical issues. In fact, a multitude of in vitro experiments are already being conducted worldwide. This chapter will introduce current in vitro experiments by dividing them into categories based on cells, culture media, and evaluation items. Finally, we will discuss a specific problem of current in vitro evaluation methods that has emerged.

### 4.1. Types of Cells

Bone is composed of cellular and extracellular matrix components. The extracellular matrix is also referred to as the bone matrix and contains a large amount of HA, which is a crystal of CaP helping to maintain bone strength. The bone matrix includes other proteins and mucopolysaccharides, notably the biopolymer type I collagen. Regarding the cellular components of bone, osteoblasts with osteogenic activity exist on the surface of the bone matrix, with osteocytes scattered inside. The cell surface of osteoblasts has high alkaline phosphatase (ALP) activity, which plays an important role in bone formation. Bone tissue is constantly being replaced by the process of remodeling, i.e., the destruction of old bone, or bone resorption, followed by bone formation. This can be referred to as endogenous bone regeneration capacity.

There are two modes of bone formation in humans: endochondral ossification and membranous ossification. In endochondral ossification, cartilage tissue is formed first, and then blood vessels enter the cartilage to be eventually replaced by bone tissue. In membranous ossification, undifferentiated cells differentiate into osteoblasts, and bone formation occurs without the formation of cartilage.

#### 4.1.1. Osteoblasts

Osteoblasts are the most important players in bone formation. After secreting bone matrix, some osteoblasts further differentiate into osteocytes, which reside in interstitial spaces called ossicles in the bone matrix. Osteoblasts differentiate from mesenchymal stem cells under certain conditions through pathways that are both structural and biochemical. For example, runt-related transcription factor 2 (Runx2) is required for the differentiation of pre-osteoblasts. Runx2 activates osteoblast-specific genes such as type I collagen, ALP, osteopontin, and osteocalcin [57]. The lifespan of osteoblasts ranges from a few days to 100 days [58]. At the end of their life, osteoblasts either differentiate into osteocytes for incorporation into new bone, become inactive bone-lining cells to protect the inactive bone surface, or die by apoptosis [59].

Various osteoblast-like cell lines have been used for the study of osteoblasts in vitro [60]. MC3T3-E1 cells were established from the cranial cortex of neonatal C57BL/6 mice by Kodama et al. in 1981 [61]. Osteoblasts form calcium-deposited bone nodules by increasing ALP and osteocalcin over time, depending on the degree of differentiation [62]. MC3T3-E1 cells have been found to increase ALP activity and form calcified nodules when simply cultured in confluent conditions [63], but are often cultured in a so-called osteogenic differentiation medium supplemented with ascorbic acid, β-glycerophosphate, and dexamethasone [64].

#### 4.1.2. Osteoclasts

Bone remodeling is a series of processes consisting of bone resorption by osteoclasts and bone formation by osteoblasts and osteocytes. Inhibiting the proliferation and activity of osteoclasts suppresses bone resorption, which in turn leads to the predominance of bone formation. As an osteoclast differentiation factor, receptor activator of NF-κB ligand (RANKL) stimulates macrophage progenitor cells to differentiate into mononuclear osteoclasts and further into the multi-nucleated mature osteoclasts that resorb bone by cell-to-cell fusion. RANKL signaling activates the transcription factor nuclear factor of activated T cells c1 (NFATc1), which expresses osteoclast-specific genes and determines cell fate. This differentiation mechanism involves multiple regulatory mechanisms, including such extracellular repressors such as osteoprotegerin and semaphorin3A, as well as intracellular repressors of NFATc1 [65].

#### 4.1.3. Chondrocytes

In the endochondral osteogenesis process, the step in which cartilage is replaced by bone tissue is critical. However, the origin of the osteoblasts at this time remains unknown. Zhou et al. [66] examined the origin of osteoblasts in cartilage by generating genetically modified mice lacking the perichondrium, the periosteum, and osteoblasts, in which chondrocytes were selectively fluorescently labeled. Their results showed that the trabecular bone in the cartilage was derived from chondrocytes and exhibited osteoblast traits. Bone tissue in the healing process of fractures was also shown to be of chondrocyte origin. Therefore, it is accepted that chondrocytes differentiate into osteoblasts during endochondral osteogenesis, which has been similarly reported by Yang et al. [67]. Novel bone regeneration methods utilizing chondrocytes are expected in the future.

#### 4.1.4. Pluripotent Stem Cells

Mesenchymal stem cells have long been recognized as undifferentiated cells that can transform into osteoblasts. Stem cells are simply defined as capable of dividing and proliferating into cells with exactly the same function and morphology (self-reproduction), and those proliferated cells are able to differentiate into a variety of other cells. Stem cells appearing in the early stages of organism development are called embryonic stem (ES) cells. Induced pluripotent stem (iPS) cells are produced by introducing multiple genes, proteins, or RNA into normal cells. In addition to ES cells and iPS cells, stem cells also exist in the tissues of adult organisms. The best-known examples of such are hematopoietic stem cells in bone marrow, which can become red or white blood cells. Bone marrow contains both hematopoietic stem cells and mesenchymal stem cells.

Mesenchymal stem cells can differentiate not only into bone tissue, but also into cartilage and fat. It has recently been reported that they can differentiate into neurons, vascular endothelial cells, and hepatocytes as well. However, the number of mesenchymal stem cells in bone marrow is very small [68], and therefore harvesting the cells by simple isolation or other manipulation requires a large marrow sample. More efficient ways to grow mesenchymal stem cells harvested from bone marrow are needed. The most common method to date is by seeding a Petri dish with as little as 3 mL of bone marrow from a donor and harvest the cells that adhere to and proliferate on the dish [69]. Despite the simplicity of this method, a large number of mesenchymal stem cells can be obtained from a small amount of bone marrow. Analysis of the surface antigens of these cells by flow cytometry shows negativity for markers specific to hematopoietic cells, such as CD34 and CD45, as well as the expression of CD29, CD73, CD90, and CD105, all of which are abundantly expressed in mesenchymal stem cells. Although mesenchymal stem cells are the most widely used pluripotent stem cells, iPS cells have also frequently been employed for bone grafting using nanostructured scaffolds in recent years [70].

### 4.2. Indicators for Assessing Bone Formation In Vitro

Bone tissue in living organisms contains osteoblasts, osteocytes, and bone matrix, in addition to an abundance of blood vessels and nerves. Hence, it is a complex environment with a three-dimensional structure. Current nerve and blood vessel regeneration research has progressed to the point where nerve cells themselves can be produced on a Petri dish [71]. However, it remains impossible to construct neural tissue with neural network function or blood vessels with a complete lumen structure on a Petri dish at this stage. In contrast, the induction of osteoblasts and osteocytes to deposit type I collagen and HA for the production of calcified nodules in bone matrix is considered biologically comparable to bone formation in vivo [72,73].

Scaffolding materials need to allow cells to adhere and proliferate, at which time they are evaluated with an appropriate system. The method for attaching cells to the scaffold material starts with adding a certain concentration of cell suspension to the scaffold material, and additional medium after the cells have attached approximately 2 h later, mainly since the scaffold material floats and floating cells have difficulty attaching to a scaffold simply placed into medium [74]. To determine cell adherence, quantitative DNA assays using Pico Green [75] or Hoechst 33258 [76] and the Alamar blue metabolic assay [54] are used with scanning electron microscopy and laser microscopy to examine cell adhesion morphology. For characterizing the depth of cell adhesion in the three-dimensional scaffold material, strategically sliced sections of the scaffold are considered as well [51,74].

Cultured mesenchymal stem cells differentiate into osteoblasts in the presence of ascorbic acid, β-glycerophosphate, and dexamethasone [64]. During this process, type I collagen and ALP are expressed at an early stage, followed next by the release of bone-specific osteocalcin, and finally the formation of a calcium-based bone matrix with a crystalline structure similar to that found in bone matrix in vivo. In order to confirm this, several gene expressions and metabolic products, such as ALP, osteocalcin, and type I collagen, are measured [77]. Staining with calcium-binding Alizarin Red S is also used as an indicator of bone matrix calcification [78]. Since Alizarin Red S cannot distinguish among minerals with different Ca/P ratios, Lammers et al. [79] demonstrated how to determine the mineral content of a sample using a combined method. Briefly, they employed quantitative wavelength-dispersive X-ray spectroscopy to determine if a bone-specific Ca/P ratio (1.67) was present in the sample, transmission electron microscopy to characterize HA-specific crystal growth, selected area electron diffraction analysis to analyze diffraction patterns, and Raman spectroscopy to compare the phosphate oxygen bonds of different CaP salts. These methods allow the distinction of mineralization as a product of cell death and matrix mineralization and avoid false positives for bone-specific mineralization.

## 5. Culture Medium

The previous chapter discussed the in vitro evaluation of biomaterials for bone regeneration. However, there are several key conditions that must not be overlooked in order to obtain stable and reproducible results with the aforementioned evaluation methods. The typical cells used in the assessment of the bone regenerative biomaterials outlined above are not necessarily established and maintained using the same cell culture methods, and thus require clarification. This chapter presents a detailed description of the culture media primarily used for osteoblastic cells.

### 5.1. Basal Medium

#### 5.1.1. Eagle’s Minimal Essential Medium (MEM)

Eagle formulated his medium by selectively removing components one by one. He concluded that 13 amino acids, 8 vitamins, 6 ions, glucose, and several compounds in the serum would be sufficient for cell growth. The result of his research was Basal Medium, Eagle (BME) and MEM, which have become widely used in cell culture [80,81].

#### 5.1.2. Dulbecco’s Modified Eagle’s Medium (DMEM)

Dulbecco modified Eagle’s medium to create DMEM [82], which is compatible with most cell types, including human, monkey, hamster, rat, mouse, and chicken. In his paper describing a plaque assay method for the polyoma virus, the medium’s formulation was described as “quadrupled with amino acids and vitamins in Eagle’s medium”. Dulbecco’s intention was presumably to increase the number of cultured cells and the viral count. It is also possible that he wanted to reduce the amount of serum since the presence of viral antibodies in the serum added to the culture medium would affect virus quantification.

Later, the non-essential amino acids serine and glycine, as well as pyruvate and iron, were added to the original Dulbecco’s medium. The reason for this was that, while MEM had been developed for the purpose of growing cells that had already been cultured, it was insufficient for sustaining tissues taken from animals for primary culture.

#### 5.1.3. Alpha Modified Eagle’s Minimum Essential Medium (αMEM)

Stanners et al. modified MEM in 1971 for the culture of mouse-hamster fusion cells to create αMEM [83]. In addition to the basic MEM, ascorbic acid, non-essential amino acids, deoxyribonucleosides, and ribonucleosides were included for cells with high nutritional requirements. Mesenchymal stem cells [84] and osteoblast-like MC3T3-E1 cells [85] are often cultured in this medium. Comparative experiments have been performed on αMEM and DMEM using primary osteoblasts derived from rat and mouse craniums [86].

### 5.2. Fetal Bovine Serum (FBS)

Serum is often added to culture medium as a source of proteins, nutrients, and cell growth factors. The most commonly used serum is FBS, which supports the growth of a variety of cells. Depending on the application, calf serum and neonatal bovine serum may also be used. These sera are naturally derived and vary among animals; some experiments cannot be reproduced if the serum is changed. Fetal serum, in which cells proliferate in the organism, contains many growth factors, including fibroblast growth factor, epidermal growth factor, insulin, transferrin, and platelet-derived growth factor, along with several high-molecular-weight and low-molecular-weight nutrient components [87,88]. Since human mesenchymal stem cells proliferate well in serum-free media [89], it is worth considering performing such experiments exclusively in serum-free medium to minimize the effects of individual serum variance.

### 5.3. Additives

There are several possible conditions for the differentiation of mesenchymal stem cells and osteoblast-like MC3T3-E1 cells into osteoblasts and the expression of their properties as osteoblasts, although the optimal conditions have not been clearly elucidated.

Mechanical and chemical stimuli are thought to be involved in osteoblast differentiation. Bones are constantly subjected to mechanical stimuli from the organism’s own body weight and muscle movement, which may be related to bone regeneration and remodeling. The types of stimuli that have been studied include compression, extension, gravity, and ultrasound [90].

Regarding chemical stimuli, a medium containing ascorbic acid, β-glycerophosphate, and dexamethasone has long been regarded as a “bone differentiation medium” [64]. The function of each of these components is described below.

#### 5.3.1. Ascorbic Acid

Ascorbic acid (vitamin C) modulates osteoblast differentiation by increasing the secretion of type 1 collagen. It is required as a cofactor for the enzymes that hydroxylate proline and lysine in procollagen [91]; in the absence of ascorbic acid, proline is not hydroxylated and the collagen chain cannot form a proper helical structure [87]. Therefore, the role of ascorbic acid in osteogenic differentiation is thought to be primarily for the secretion of type 1 collagen into the extracellular matrix. As mentioned above, ascorbic acid is added to αMEM. There is concern regarding the effect of this on the differentiation of MC3T3-E1 cells, and consequently they are often maintained in ascorbate-free αMEM [92,93,94]. Furthermore, ascorbic acid is very unstable, thus prompting experimental systems that substitute it with analogues [60].

#### 5.3.2. β-glycerophosphate

β-glycerophosphate acts as a phosphate source for bone mineralization and induces the expression of osteogenic genes. Recent findings have shown that inorganic phosphate, in addition to being a necessary source of phosphate for the formation of HA, is an intracellular signaling molecule that regulates osteopontin gene expression [95,96]. Moreover, β-glycerophosphate regulates the expression of BMP-2 and many osteogenesis-related genes by activating the extracellular signal-regulated kinase signaling pathway and the cyclic-AMP/proteinase-A pathway [97].

#### 5.3.3. Dexamethasone

Dexamethasone induces Runx2 expression through four and a half of LIM-only protein 2 (FHL2)/β-catenin-mediated transcriptional activation. Hamidouche et al. [98] demonstrated that a dexamethasone-induced differentiation of mesenchymal stem cells into osteoblasts was mediated by activating Runx2 expression in a WNT/β-catenin signaling-dependent manner. The binding of dexamethasone to the glucocorticoid response element in the promoter of FHL2 is thought to upregulate FHL2.

## 6. Opportunities and Challenges

### 6.1. Smart Nanomaterials

It is well known that organisms in nature can dynamically change their characteristics to adapt to their surrounding environment. Biological systems are diverse and complex, but also possess the ability to accommodate environmental changes in order to maintain normal function. In the field of medical materials, so-called “smart nanomaterials” are being developed as substances that can respond to a wide variety of stimuli by altering their characteristics, such as shape, surface area, size, permeability, solubility, and mechanical properties. These responses may or may not be reversible. In particular, polymer-based materials have been proposed as a promising option for fabricating responsive structures since their properties are easily controllable. A variety of polymers are being developed that respond to physical (temperature, light, ultrasound, electrical, magnetic, mechanical), chemical (pH, solvent, electrochemical), and biological (enzymatic) stimuli. Advanced polymer types are dual- or multiple-stimulus responsive, thus adapting to multiple stimuli simultaneously [99]. Although smart nanomaterials have not yet been widely employed for bone regeneration, they have already been applied in wound healing and drug delivery. Table 3 shows examples of smart nanomaterials and their biomedical applications [99].

### 6.2. Effects of Medium and Additives on MC3T3-E1 Cells: An In Vitro Risk

We recently reported on the potential problems in existing in vitro evaluation methods of bone materials, whereby different media conditions could lead to inconsistent results, by means of MC3T3-E1 cells, which have long been employed in bone regeneration and osteogenesis research [11]. We cultured MC3T3-E1 cells in αMEM and DMEM, which are the basic media for osteoblast experiments, as well as in αMEM without ascorbic acid and in a conditioned medium consisting of αMEM without ascorbic acid and DMEM with the subsequent addition of ascorbic acid, and compared several indicators of osteogenic potential, including cell proliferation, ALP activity, osteoblast marker expression, and calcification (Figure 4). Our results show that the factors for evaluating MC3T3-E1 cell osteogenic potential differed greatly depending on the culture medium used, and that basic medium conditions, such as the type of medium, the presence of ascorbic acid, and medium freshness, might profoundly alter the bone regeneration potential of biomaterials. In other words, such differences in medium conditions may have a stark effect on osteoblastic cells, and experimental findings can differ even when the biomaterial is the same. This result is alarming with regards to the current situation, in that it is difficult to make objective comparisons due to the diversity of experimental methods used for assessing osteogenic functions. In the field of cell therapy, consensus on serum substitutes and serum-free media is becoming increasingly important due to safety issues and the aforementioned problems with serum variability that affect the reproducibility of the final product [105,106]. Even in bone regeneration research, we support greater consensus and unity among research groups to provide more standardized methodologies that are readily applicable worldwide.

## 7. Conclusions

Today, many protocols are being used in bone regenerative medicine involving nanomaterials [10]. As the mechanism of bone regeneration is not completely clear, it is impossible to reproduce this process, and all protocols must be said to represent only a part of actual bone regeneration. It is important to know under exactly what culture conditions osteoblasts change in in vitro experimental systems. For instance, the degree of MC3T3-E1 cell ossification varies greatly in different cell culture media [11], and there is a risk of drawing false conclusions if the results of multiple studies conducted under different media conditions are evaluated in the same manner. Based on this premise, it will be necessary to evaluate each experiment with an awareness of what principle and what part of bone regeneration is being reproduced. In order to conduct reproducible experiments, it will be paramount to faithfully replicate previous experimental systems, set up appropriate controls, and conduct objective evaluations without bias, overestimation, or underestimation. Researchers need to think critically about the purpose of their experiment and select an appropriate experimental system in the context of similar research. From the standpoint of animal welfare, a careful review of the many in vitro evaluation methods to date is required in order to identify those best reflecting the conditions in the living body from both a humane and a scientific standpoint. By aligning research conditions and goals, a more unified and presumably effective approach to bone regenerative medicine using nanomaterials can be achieved.

## Figures and Tables

**Figure 1 nanomaterials-12-01195-f001:**
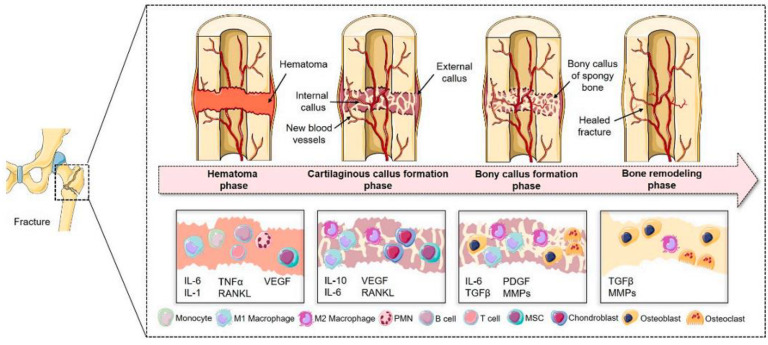
Schematic of the four phases of fracture healing. Reprinted from reference [23]. IL-1, interleukin 1; IL-6, interleukin 6; IL-10, interleukin 10; TGF-α, β, transforming growth factor-alpha, beta; RANKL, receptor activator of nuclear factor-κB ligand; VEGF, vascular endothelial growth factor; PDGF, platelet-derived growth factor; PMN, polymorphonuclear leukocyte; MMP, matrix metalloproteinase.

**Figure 2 nanomaterials-12-01195-f002:**
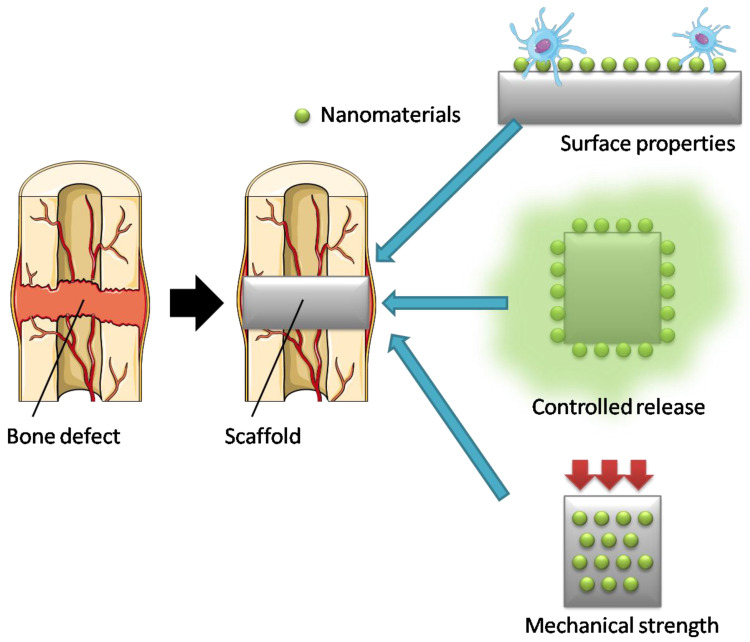
Schematic of nanomaterials used for bone tissue regeneration. Figure was modified from Servier Medical Art, licensed under a Creative Common Attribution 3.0 Generic License.

**Figure 3 nanomaterials-12-01195-f003:**
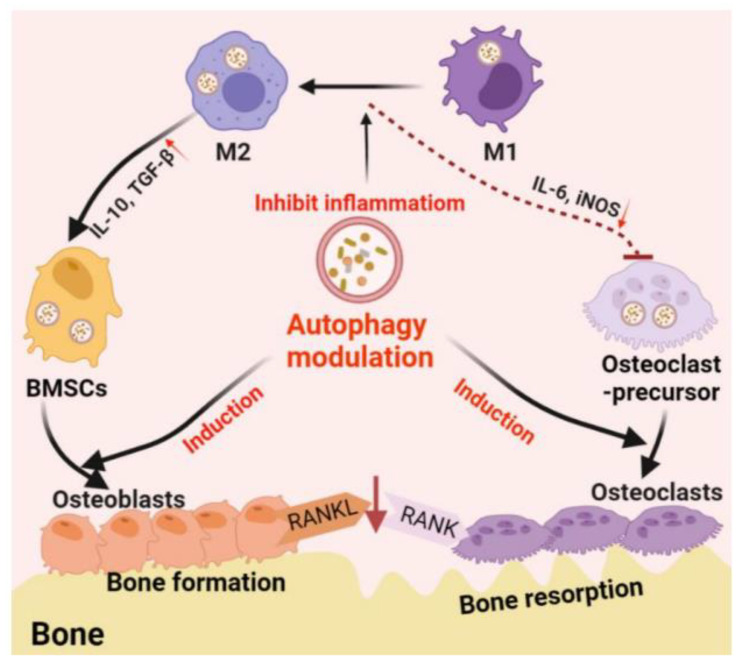
Schematic of autophagy-derived regulation in the differentiation/function of osteoclasts/osteoblasts and osteoimmunology. Reprinted from reference [36]. M1, M1 macrophage; M2, M2 macrophage; IL-6, interleukin 6; iNOS, inducible nitric oxide synthase; IL-10, interleukin 10; TGF-β, transforming growth factor-beta; RANKL, receptor activator of nuclear factor-κB ligand; BMSC, bone marrow stem cell.

**Figure 4 nanomaterials-12-01195-f004:**
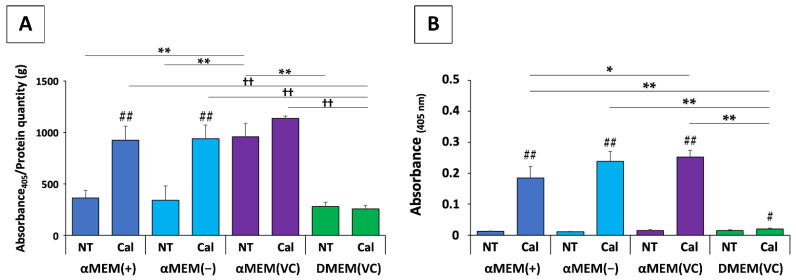
Example of evaluations that differed remarkably depending on the medium used. (**A**) Quantitative analysis of alkaline phosphatase activity (** *p* < 0.01, ^††^ *p* < 0.01, ^##^ *p* < 0.01). ** multiple group comparisons among nontreated (NT) media; ^††^ multiple group comparisons among calcification (Cal) media; ^##^ two-group comparisons between NT and Cal media. (**B**) Quantitative analysis of Alizarin Red S staining (* *p* < 0.05, ** *p* < 0.01, ^#^ *p* < 0.05, ^##^ *p* < 0.01). *^,^ ** multiple group comparisons among calcification (Cal) media; ^#, ##^ two-group comparisons between nontreated (NT) and Cal media. Image is modified from a study by Izumiya et al. [11].

**Table 1 nanomaterials-12-01195-t001:** Advantages and drawbacks of inorganic nanomaterials.

Nanomaterial	Advantages	Drawbacks
Ceramics	Biocompatibility Osteoinductive potential	Potential for cytotoxicity
Polymers	Biocompatibility Biodegradability Manufacturing flexibility	Unfavorable biodegradability
Carbon	Mechanical strength Electrical conductivity	Non-degradability Concerns of long-term safety
Gold	Biocompatibility Photothermal stability Near-infrared absorbance	Non-degradability Concerns of long-term safety
Titanium-based nanomaterials	Load-bearing properties Biocompatibility	Non-degradability Poor biological response and anti-bacterial properties
Liposomes	Drug-loading ability	Mechanical weakness

**Table 3 nanomaterials-12-01195-t003:** Examples of smart nanomaterials and their biomedical applications. Modified from reference [99].

Stimulus	Nanomaterial	Application	Reference
Temperature	Gold nanoparticles—Pluronic^®^F127- Hydroxypropyl methylcellulose	Tissue engineering	[100]
pH	Polyethylene glycol-Ag nanoparticle	Antibacterial, wound healing	[101]
Redox	Prodrug/AgNPs hybrid nanoparticles	Drug delivery	[102]
Glucose	Boronic acid-derived polymers	Drug delivery	[103]
Enzyme	Layer-by-layer assembly of poly (2-oxazoline)-based materials	Therapeutic delivery	[104]

## Data Availability

This is a review article and the information is available in literature.

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
