# Peer review of "Current Methods in the Study of Nanomaterials for Bone Regeneration"

_nanomaterials, 2022, doi:10.3390/nano12071195_

Round 1

Reviewer 1 Report

  1. Authors should provide a schematic diagram of autophagy-derived regulation on the differentiation and function of osteoclast, osteoblast, and osteoimmunology
  2. Authors should provide tables and diagrams of the main inorganic nanomaterials used for bone tissue regeneration
  3. Table for the advantages and drawbacks of inorganic nanomaterials.
  4. Explain bone fracture repair processes
  5. Should include an explanation about smart nanomaterials
  6. Include other nanomaterials like polymeric composites, nanotubes, nanofibers etc.
  7. Should explain the chemical and physical properties of nanomaterials
  8. Provide a table for engineering nanomaterials for bone regeneration
  9. Should include challenges and opportunities
  10. Review article should contain at least 3-4 figures and tables with a comparison of previously reported data

Reviewer 2 Report

This reviews deals with current methods in study of nanomaterials for bone regeneration. This is a very interesting topic and should be explored. 

there are several issues with the current form of this paper:

1) it is not clear why this review paper is unique and offer different point of views. The authors should have made it very clear in the text. 

2. the paper reads like a protocol, with very detailed information about cell types and cell culture media components. I do not see the need for this information. 

3. The paper discussed more on the scaffold than bone regeneration. I am not sure if this focus shift is wise if the title is "study of nanomaterials for bone regeneration". 

English is easy to read and helps to follow the text. 

Round 2

Reviewer 1 Report

Accept in present form